# Patient, parent and provider perspectives on sickle cell disease genetics research in Jamaica

Krystin Jones[1,2]*, Kearsley Stewart[1], Monika Asnani[3], Charmaine D. Royal[1,4,5]

**1** Duke Global Health Institute, Duke University, Durham, North Carolina, United States of America, **2** Department of Epidemiology, Bloomberg School of Public Health, Johns Hopkins University, Baltimore, Maryland, United States of America, **3** Caribbean Institute for Health Research-Sickle Cell Unit, The University of the West Indies, Mona Campus, Kingston, Jamaica, **4** Department of African & African American Studies, Duke University, Durham, North Carolina, United States of America, **5** Center on Genomics, Race, Identity, Difference, Duke University, Durham, North Carolina, United States of America

* krystin.amoi2@gmail.com

## Abstract

Advances in genetics and genomics research are revolutionizing the way we understand sickle cell disease (SCD) and approach its treatment and management. Much of this research has been conducted in high-income countries and so much of the available data is skewed towards these populations. Efforts are now being made to facilitate this research in low-and-middle-income countries (LMICs) through capacity building and strengthening. These efforts must also include understanding context specific SCD stakeholder perspectives and attitudes to guide genetics and genomics research in these populations. This qualitative study used semi-structured in-depth interviews to investigate perspectives on SCD genetics research among 10 SCD healthcare providers, 10 individuals living with SCD and 9 parents affiliated with the Sickle Cell Unit (SCU) in Kingston, Jamaica. Most participants showed support for or a willingness to participate in SCD genetics research as they believed it would lead to improved SCD treatment options and greater knowledge about the disease. Some patients and parents, however, seemed to conflate genetics research participation with receiving SCD treatment or health screenings, pointing to therapeutic and diagnostic misconceptions. Skepticism about genetics research also emerged among some participants due to concerns about loss of privacy, mistrust and fears of misuse, the potential time commitment and inconvenience, and fear of the unknown. Overall, Jamaican SCD stakeholders conveyed an openness towards SCD genetics research participation. However, researchers in genetics must be mindful of the power imbalances that exist between researchers and research participants in LMICs. Steps must be taken to ensure that communities in LMICs are not only able to make contributions to genetic studies, but also that they understand the research goals and the implications of their participation. Working alongside local researchers, providers, patients, and other interested parties will be a key element of facilitating trustworthy and impactful research and establishing trust within these communities.

**Data availability statement:** All relevant data are within the paper and its Supporting information files.

**Funding:** This work was supported by the Global Health Institute at Duke University (KJ). The funders had no role in study design, data collection and analysis, decision to publish, or preparation of the manuscript.

**Competing interests:** The authors have declared that no competing interests exist.

## Introduction

Sickle cell disease (SCD) is the most common genetic disorder of the blood. Globally, over 300,000 children are born each year with SCD and models suggest that this number could rise to 400,000 by 2050 [1,2]. Approximately 95% of people with SCD live in low- and middle-income countries (LMICs) including India and countries in Africa and the Caribbean [1]. Outside of West Africa, the Caribbean, which benefits from long-standing prenatal and newborn screening programs across several islands, has the highest SCD incidence [1,3].

SCD is one of the leading causes of death due to non-communicable disease worldwide. Vaso-occlusive crises are the hallmark of the disease in which the abnormal sickle hemoglobin (HbS) distorts the shape of the red blood cells resulting in blockage of blood vessels, leading to various acute complications [4]. Though these acute events largely define the pathophysiology of the disease, repeated vaso-occlusion can result in chronic parenchymal damage to almost every organ system in the body and shortened life expectancy [1,2,4,5]. In addition, individuals living with the disease worldwide face psychosocial issues such as stigma, depression and loneliness, and poor quality of life, as well as wide-scale socioeconomic insecurity [4,6–8].

SCD features significant phenotypic variation that belies its simple monogenic etiology [2,4]. Genetic and environmental factors, and their interaction are believed to play a role in this variability [1,8]. Genetics and genomics research have been essential in defining the nature and course of SCD. Advances in these fields are now revolutionizing how we understand SCD complexity and how we approach SCD treatment and management, including the development and recent approval of CRISPR-based gene editing therapies [9,10]. However, genetics research in general, and SCD research in particular, has been generally limited to high-income countries (HICs). Current estimates indicate that a large, and increasing, proportion of genetics studies have been conducted among individuals with recent genetic and genealogical ancestors from Europe [11]. On the other hand, a mere 1.1% of individuals in these studies were of recent African ancestry, the majority of which were from the United States (US) or the United Kingdom (UK), representing only a subset of African genetic diversity [11]. As a result, available genetics-related health data are largely skewed towards higher income regions and populations of European genetic and genealogic ancestry. The transportability of these genetic data to other populations is therefore limited, bearing negative consequences for scientific advancement and health equity [11,12]. Much of these disparities in genetics research participation largely stem from under-equipped research and health infrastructures in LMICs [13].

Research consortia such as the Human Heredity and Health in Africa (H3Africa) initiative have advanced work to develop genetics and genomics research infrastructure, including biorepositories, throughout the African continent to investigate diseases including SCD [13]. Genomics research is emerging more slowly in the Caribbean. The region's significant genetic diversity and admixed genetic ancestry from European, African, and Indigenous ancestral populations, however, may offer unique insights into genetic contributions to disease variability, making it an ideal target for genetics research in general and SCD genetics research in particular [14–17].

However, successfully exploiting genetics research to better understand SCD pathophysiology and further facilitate treatment development largely depends on participant willingness to enroll and remain in studies. Factors including imperfect understanding of genetic concepts and research processes, and mistrust in researchers due to historical violence and injustice including exploitation can impede research participation, particularly among individuals in LMICs [11]. Therefore, in order to facilitate effective research within these populations, it is fundamental that researchers understand stakeholders' perspectives on genetics research, and motivators and deterrents to genetics research participation.

Quantitative and qualitative work to understand drivers of and barriers to genetics research participation has been conducted in various communities and for various diseases globally, though much of this has been largely concentrated in HICs [18–20]. Studies engaging immigrant populations in the US have also provided important insight, highlighting issues of colonial mistreatment and exploitation as barriers to genetics research participation [21]. However, few studies have engaged SCD communities in the US and Africa on genetics research participation [22–24]. To our knowledge, no studies have similarly engaged SCD stakeholders in Jamaica or the wider Caribbean.

In the present study, we examined SCD patient, parent, and provider perspectives on SCD genetics research in Jamaica, the third largest island in the Caribbean. Jamaica, where the coverage of newborn screening (NBS) programs exceeds 99.9%, has the highest recorded SCD prevalence (0.65% or 1 in 153 live births [1,3]) in the region. This high SCD prevalence and the significant contributions Jamaica has made to SCD research have made the island an ideal site for these investigations. Therefore, we explored the understanding of SCD genetics research and motivators and deterrents to genetic research participation among these stakeholders. Using these perspectives, researchers may develop a potential framework to guide the conduct of genetics research in Jamaica. Additionally, this research may inform further stakeholder engagement efforts in the wider Caribbean.

## Methods

This qualitative study was a part of a larger study that engaged SCD stakeholders in Jamaica about their perspectives on genetics research and curative therapies for SCD. Ethics approval was obtained from the Campus IRB at Duke University [2021–0014] and the Ethics Committee at the University of the West Indies, Mona [ECP 178, 19/20].

### Participants and recruitment

Participants were recruited from the Sickle Cell Unit (SCU). The SCU is part of the Caribbean Institute for Health Research. Participants were ten (10) SCD patients who were 18 years or older, 9 parents of pediatric SCD patients and 10 healthcare providers who worked at the SCU in clinical care or research. From September to December 2020, Providers were recruited through convenience sampling, while patients and parents were randomly selected from a list of patient data. However, given the greater number of female patients and parents being enrolled into the study, we then implemented a purposive sampling strategy to ensure that male patients and parents were included. Sample sizes were chosen with the aim of achieving thematic saturation within each group and across the entire study population [25,26].

### Data collection

Data was collected through individual in-depth interviews to facilitate interviewee comfort and gain deeper, individual insights on personal or sensitive topics including experiences of stigma and other challenges associated with SCD. Interviews were conducted by the first author (KJ) between September and December 2020. At the start of each interview, the interviewer (KJ) obtained verbal informed consent from each participant witnessed by an SCU staff member. Due to Covid, interviews were conducted remotely, using the videoconferencing platform, Zoom. They lasted between 45 and 90 minutes and were conducted in a mixture of English and Jamaican dialect (i.e., Patois). Interviews were recorded using Zoom.

## Data collection instruments

The instrument used in this study was a semi-structured in-depth interview guide (S1 Appendix). The guide was used to ensure that interviews were standardized. The interview guide adapted and modified the community engagement format and surveys developed by the Sickle Cell Disease Genomics Network in Africa (SickleGenAfrica) [22]. Previously used in Ghana (Accra), Nigeria (Abuja, Kano and Lagos), and Tanzania (Dar es Salaam), these instruments were developed to explore specific issues in genetics research, including sample collection and storage in biorepositories [22]. The instruments were modified into one document to fit the semi- structured, in-depth interview format, and to include questions that explored issues surrounding curative therapies for sickle cell disease. Further questions explored knowledge of and attitudes towards SCD cures, individual experiences with SCD management and treatment, and the socioeconomic and psychosocial challenges related to SCD to better contextualize participant attitudes toward genetics research and SCD cures.

## Analysis

Transcripts were produced by the Zoom audio transcription function. The transcripts were then cleaned and edited for accuracy. Words or phrases in Jamaican Patois were rewritten into Standard English, or otherwise clarified where it was determined necessary and where doing so would not interfere with the sentiments or ideas the participant sought to express. Participant demographic survey information was compiled into an Excel document. These documents were imported into NVivo 12 for analysis. Transcripts were closely read and structurally coded by a single coder (KJ) based on interview questions and the general structure of the interview guide (S2 Appendix) through thematic analysis based on the methods described in Saldaña [27]. Codes and themes were reviewed by a second author (CR). Themes related to participants' attitudes towards genetics research were identified. These themes and corresponding codes were compiled into a master codebook that defined codes and specified their potential uses. Interview transcripts were then further coded based on these emergent themes. Memos were written to summarize preliminary findings from this first pass thematic coding [27]. Additional themes identified during these processes were added to the codebook and transcripts were recoded where necessary. Nvivo queries, including matrix coding and cross tabulation, were then run to compare codes across participant types and to make comparisons between relevant codes.

## Results

A total of 29 interviews were conducted including 10 Providers, 10 patients and 9 parents. Participant demographic information is summarized in Table 1. Participant age ranged from 22 to 64 years. Providers had the highest average age of 44.6 years followed by parents (38.1 years) and patients (36.1 years). Approximately seventy-two percent of respondents were female. Only 2 male providers, 3 male patients and 3 male parents participated in the study. Approximately 82% of participants identified as black and 93.1% identified as Christian. While most providers and patients (100% and 60% respectively) had some tertiary level education, most parents' (88.9%) highest education level was at the secondary level. Thirty percent (30%) of providers, 100% of patients and 55.6% of parents had a family member (other than themselves or their children) with SCD. Providers were either doctors, nurses, or social workers. Patients' reported genotypes included SS, SC and Sß-Thalassemia, and a range of illness severities. Finally, parents included 6 mothers, 2 fathers, 1 grandmother, and 1 uncle who were either the child's primary caregiver, or co-caregiver.

Below, we present our findings regarding participants' perspectives on individuals with SCD participating in SCD genetics research. We identified two overarching themes from these perspectives:

1) Excitement and enthusiasm

2) Skepticism and reluctance

**Table 1. Participant Characteristics.**

| | Patient | Parent | Provider | Total |
|---|---|---|---|---|
| | (n = 10) | (n = 9) | (n = 10) | (N = 29) |
| *Age (in years)* | | | | |
| *Mean (SD)* | 36.1 (10.9) | 38.1 (5.80) | 44.6 (14.2) | 39.7 (11.3) |
| *Median [IQR]* | 35.0 [29.0, 43.0] | 35.0 [31.0, 46.0] | 43.0 [30.0, 59.0] | 35.0 [31.0, 46.0] |
| *Self-Identified Race/Ethnicity* | | | | |
| *Black* | 8 (80.0%) | 9 (100%) | 7 (70.0%) | 24 (82.8%) |
| *Mixed* | 2 (20.0%) | 0 (0%) | 3 (30.0%) | 5 (17.2%) |
| *Sex* | | | | |
| *Female* | 7 (70.0%) | 6 (66.7%) | 8 (80.0%) | 21 (72.4%) |
| *Male* | 3 (30.0%) | 3 (33.3%) | 2 (20.0%) | 8 (27.6%) |
| *Highest Education Level* | | | | |
| *Primary* | 1 (10.0%) | 0 (0%) | 0 (0%) | 1 (3.4%) |
| *Secondary* | 3 (30.0%) | 8 (88.9%) | 0 (0%) | 11 (37.9%) |
| *Tertiary* | 6 (60.0%) | 1 (11.1%) | 10 (100%) | 17 (58.6%) |
| *Employment* | | | | |
| *Employed* | 9 (90.0%) | 5 (55.6%) | 10 (100%) | 24 (82.8%) |
| *Unemployed* | 1 (10.0%) | 4 (44.4%) | 0 (0%) | 5 (17.2%) |
| *Religion* | | | | |
| *Christianity* | 10 (100%) | 8 (88.9%) | 9 (90.0%) | 27 (93.1%) |
| *Other* | 0 (0%) | 1 (11.1%) | 1 (10.0%) | 2 (6.9%) |
| *SCD in Family[1]* | | | | |
| *Yes* | 10 (100%) | 5 (55.6%) | 3 (30.0%) | 18 (62.1%) |
| *No* | 0 (0%) | 4 (44.4%) | 7 (70.0%) | 11 (37.9%) |

## 1) Excitement and Enthusiasm

Participants were generally enthusiastic about SCD genetics research as they believed it could a) lead to better treatment options or b) advance SCD knowledge. This theme is summarized in Table 2 and further explained below.

### a) Better Treatment Options

Providers, patients, and parents believed genetics research would bring about improved SCD treatment options (Table 2). Providers discussed the physical, mental, and emotional challenges SCD patients and their parents face. They explained that patients and parents were desperate for a solution and often found the options available to them limited.

**Table 2. Excitement and Enthusiasm Coding Matrix Breaks Down Number of Participant That Endorsed Each Identified Motivators for SCD Genetics Research Participation by Participant Type.**

| | Patient n = 10 | Parent n = 9 | Provider n = 10 | Total n = 29 |
|---|---|---|---|---|
| *Better Options and a Better Life* | | | | |
| *For Themselves* | 4 | 4 | 1 | 9 |
| *For Others* | 8 | 2 | 1 | 11 |
| *Advance Knowledge* | | | | |
| *Medical* | 2 | 0 | 5 | 7 |
| *Personal* | 6 | 4 | 0 | 10 |

"The emotional and mental despair would really lead them to, to make that decision too. It's not just the physical. They have to just reach to a point where they say, you know, I just cannot live, I don't want to live like this anymore. I will-I'm willing to try something else." (Provider, female, 50's)

Some patients revealed that they were dissatisfied with current SCD treatment options, finding them ineffective or having grown tired of taking daily medications. As a result, respondents were willing to support or participate in SCD genetics research with the of hope of developing better treatment options or cures for SCD patients that would lessen many of these challenges. One patient explained,

"Just taking these medications every day. I will be willing, more than willing to be in this research. Because I don't like to be taking medications every day. And it's not one time, it's more than one time for the day."
(Patient, Female, 20's)

Another patient stated,

"…most people such as myself, want to get rid of it so, many sickle cell patients will try anything just to see if there's a cure out there to get rid of it or just to minimize the effects of this disease. So, basically, many of them will try because [of] what they had borne in the past, or right now, if they can get a relief from that or basically control it, they will jump at the chance to." (Patient, Male, 20's)

Patients and parents considered the relief that participating in genetics research would bring them or their child, and some seemed to conflate participating in genetics research with receiving treatment or a cure for the disease. However, many patients also displayed altruistic motivations and hoped that by participating in genetics research studies they could help improve the lives of others living with the disease. One patient stated,

"The fact that I'm helping others…as I said before, I love helping persons. And I don't want to see persons suffering as much as I did. So, for me to participate and to help others, I am willing to do that." (Patient, Female, 40's)

**b) Advancing SCD Knowledge**

To a lesser extent, respondents' support for or willingness to participate in SCD genetics research was related to the belief that it would advance SCD knowledge (Table 2). Providers explained that SCD genetics research would lead to improved understanding of the disease pathophysiology and symptoms. One provider, for example, discussed a study he was involved in that planned to use genetic research to better understand SCD pain:

"I just recently submitted something that requests subjects to do genetic testing. And that's because… it(genetics) is very important to understand certain things about pain and so on." (Physician, Male, 30's)

Providers also voiced their support for Jamaican SCD patients specifically participating in genetics research as they believed this would create a wider and more inclusive knowledge base. One provider explained,

"If we can say this is our experience and this is what we found that works or this is how our patients are unique, and we can contribute to that. And it might be that what is unique to us is also unique to another subset of patients somewhere else. And it might be something that they're experiencing that's different from others in either the first world countries or places that are more temperate or whatever the differences that might be that are identified."
(Physician, Female, 30's)

Patients also reflected on the gaps present in SCD knowledge and saw SCD genetics research as a way to help fill these gaps. They praised the advances that had been made recently but felt that current knowledge was inadequate and there was still much to be understood. However, patients and parents of patients more often saw SCD genetics research as a way to better understand their personal health or ancestry (Table 2). They viewed it as a sort of health screening that could predict future health issues or provide guidance on how to better care for one's health. One patient stated,

> "I want to be able to do it and if they should find something harmful or dangerous, or if you they would want to warn me about something that may arise in the future may not, so, I would want to know." (Patient, Male, 20's)

Other parents saw in it a way to better understand heredity and, in some cases, prevent the disease in future generations through more informed partner selection. One parent explained,

> "Well, I guess I guess because like my son that he is the first and I don't hear of any other family members that have it, you know, he would want to trace it to find out, you know, yeah." (Parent, Female, 40's)

While another stated,

> "…they want to find a cure or know where it is coming from. So, you know, you have to choose their partner and know who they're choosing if that person has sickle cell then you know that they cannot partner with that person because they will bring forth a sickle cell baby." (Parent, Female, 40's)

**2) Skepticism and Reluctance**

Participants also displayed skepticism or reluctance towards SCD genetics research participation. This was driven by concerns about a) invasiveness, b) misuse, c) the unknown, and d) the time and inconvenience. This theme is summarized in Table 3 and explained in the sections that follow.

**a) Invasiveness or Loss of Privacy**

Respondents discussed concerns about the invasiveness or loss of privacy associated with genetics research as a source of hesitancy. Patients and parents in particular felt that details about family and health were private matters and worried that participating in SCD genetics research would expose them. Providers also recognized these concerns among patients and parents, and also named them as a source of reluctance. One provider explained

> "…people don't want them to have access to their private information and they're not sure what the researchers are going to do. It's going to always be your DNA, [it's] just your DNA. It's just one you. So, it will always be linked back to you. So that is always a reservation that the patients will have and the parents of the children who do the consent for

**Table 3. Skepticism & Reluctance Coding Matrix Breaks Down Number of Participant That Endorsed Each Identified Deterrent for SCD Genetics Research Participation by Participant Type.**

|  | Patient n = 10 | Parent n = 9 | Provider n = 10 | Total n = 29 |
|---|---|---|---|---|
| *Invasiveness* | 7 | 7 | 5 | 19 |
| *Mistrust* | 4 | 3 | 7 | 14 |
| *The Unknown* | 2 | 2 | 3 | 7 |
| *Time & Inconvenience* | 1 | 3 | 4 | 8 |

them. I mean, we will always have, "how can you guarantee that my, my child's information will not be leaked, or my information will not be leaked?"" (Nurse, Female, 40's)

Most patients and parents also mentioned fears about the physical invasiveness of genetics research as a deterrent to participation. This emerged particularly during discussions about the types of samples that would be donated for research, especially among parents of younger children:

"…if it's a case where there's a lot of needle poking and stuff and, and things like that, I'm not going to subject my child to that. So, anything that has to, asks my child to go to any form of pain or stuff like that, I'm not going to subject my child to that. It would have to be a case where I have no other alternative." (Parent, Female, 30's)

**b) Mistrust and Fears of Misuse**

Hesitancy around participating in SCD genetics research also stemmed from mistrust of researchers and fears of misuse. Respondents described their own feelings of mistrust and those they heard from members of their communities. They expressed concerns about the unethical use of patients' genetic information. Patients and parents also worried that researchers would use their or their child's information for reasons not previously agreed upon:

"You can't trust these people… Because suppose they're storing it and um, they go and use it, they pinch a piece of it for their own purpose." (Patient, Female, 50's)

As a result, some patients and parents explained that they only felt comfortable if researchers and providers they trusted at the SCU had access to their genetic information:

"I think, I don't want it to be out in the wild. I just, like how I know the sickle cell clinic, is just them I would want to know about that." (Patient, Female, 50's)

Respondents explained that misinformation, conspiracy theories and religious beliefs sometimes drove negative perceptions of research and researchers in general and fostered feelings of mistrust. One provider explained,

"You know, they believe that, you know, people have all kind of theories that you put some chip in them body, that you going control their mind, you know…all kinds of conspiracy things." (Nurse, Female, 50's)

Providers also explained that much of this mistrust was founded in the long history of misuse and unethical practice in health research:

"I think that, like I said, there is there is an ugly history of misuse of things. And sometimes it's, it's seemingly benign. You know, it's just scientific excitement and fervor that drives these things. And sometimes it's something more malicious than that. But, but there's, there are examples of these things being misused or used in ways that that were anticipated and not agreed to previously and, and it's important to catch these." (Physician, Female, 40's)

**c) Uncertainty and Fear of The Unknown**

Uncertainty or fear of the unknown was another source of participants' skepticism towards SCD genetics research. Providers explained that since such research was not commonplace, patients and parents who had limited knowledge of the process and could not turn to their peers for information or advice would likely hesitate to take part. One provider explained,

> "It's difficult to sit beside somebody in the clinic and get somebody's experience because not a lot of people have had this done, so I think the fear of the unknown may stop some people you know from participating." (Nurse, Female, 40's)

Patients and parents also explained that for some, a fear of knowing or learning more about their health would be a deterrent to SCD genetics research participation:

> "People afraid of things that them don't know. Cause you have some people they're not going to take a sickle cell test because they don't want to hear that they have sickle cell." (Parent, Female, 40's)

**d) Time and Inconvenience**

Finally, but least often, providers, patients and parents noted that the time and inconveniences involved with genetics research participation were other sources of hesitancy for SCD patients and their parents. They explained that those unreceptive to research may simply be those who either were unable to dedicate the required time and energy or were hindered by their location. Some parents and patients worried about the burden of this commitment and were unenthused about the research they felt was too involved or too great an inconvenience. One parent explained,

> "If it's the case that it's going back and forth with this, it doesn't make any sense." (Parent, Female, 40's)

One provider reflected on the fact that, as genetic studies often took place overseas, they tended to be inaccessible for Jamaican participants and those that may have wanted to participate were simply unable to:

> "They're going to have to do a lot of traveling and all of that is a huge thing to consider. So as far as I know, the persons who have done it, have been persons who had you know living or monetary situation that they could accommodate all of that." (Physician, Male, 30's)

## Discussion

Previous studies in the US and the UK have examined stakeholder perspectives on SCD genetics research, and H3Africa and other research groups are beginning to examine these perspectives in some African countries. However, to our knowledge, this is first study that has explored these issues in Jamaica or the wider Caribbean.

Participants discussed several drivers of and barriers to participation in SCD genetics research. Providers, patients and parents believed that genetics research participation would result in improved SCD treatment for SCD patients. Many participants understood that genetics research could yield a greater understanding of SCD and thereby result in the development of better options for SCD treatment and management, such as SCD cures. In this regard, many patients also expressed altruistic motivations for genetics research participation. They hoped their participation in genetics research would create tools for improved SCD management not only for themselves, but for the wider SCD population.

All participant groups also hoped genetics research would improve the overall SCD knowledge base. Providers emphasized the importance of Jamaicans' participation in such research towards achieving this goal. However, many patients' and parents' interest in genetics research was driven by the personal SCD knowledge they believed it would provide. Patients and parents felt that genetics research would provide a means by which certain health concerns could be diagnosed or better defined, and would offer insights into overall health and heredity. They also believed it would empower SCD patients to make informed decisions about partner selection and reproduction, and to ensure that their children and grandchildren did not face the same challenges they had. These attitudes towards SCD knowledge were similar to those seen within other communities. Previous studies found that Kenyan SCD stakeholders and Black African immigrant

community leaders spoke of the benefits of acquiring personal SCD knowledge through research [21,28]. Likewise, they believed this knowledge would empower members of the SCD community to make informed health decisions and be better able to "manage future reproductive risks" [21,28].

Patients' and parents' belief that research participation would yield personalized results about heredity and individual health pointed to what Tindana and de Vries termed diagnostic misconception [29]. They defined this as a phenomenon in which participants conflated research with health screening or diagnostic efforts [29]. Some patients and parents also conflated SCD research with receiving SCD treatment. This therapeutic misconception became increasingly evident as many patients and parents discussed genetics research participation as if speaking about receiving SCD treatments or cures. They appeared to view these two concepts as the same, rather than seeing genetics research as a path to improved SCD treatment options. Many patients and parents therefore framed patient participation in genetics research around this therapeutic misconception.

Though evident in many settings, both therapeutic and diagnostic misconception have been widely described in relation to studies conducted in LMICs [29–32]. Tindana, et al., for example, described this phenomenon in participant understandings of the MalariaGEN study and their motivations for study participation [30]. The presence of these misconceptions among certain patients and parents in that study suggested limited understanding of the aims and methods involved in genetics research, similar to what was observed in this present investigation. Therapeutic and diagnostic misconceptions could have significant implications for participant autonomy. They can create false inducement among study participants and result in confusion or unmet expectations [32]. Furthermore, participants who carry these misconceptions may not understand the implications of their consent. Particularly, confusion may arise regarding key elements of genetics research studies such as biospecimen storage and future sample use [29].

Despite this generally positive attitude towards genetics research, respondents also expressed reservations about SCD genetics research participation. Mistrust, and invasiveness and loss of privacy were significant points of concern among providers, patients, and parents. Mistrust was most often described by healthcare providers. Providers, patients, and parents described both their own concerns about misuse and mistrust, and concerns heard within their communities. Mistrust and fears of misuse often stemmed from misinformation about researcher motivations and spread by way of online sources, or throughout communities by word of mouth. Some participants, however, cited a history of abuse and misconduct in research that disproportionately affected black and other minoritized peoples globally [33,34].

Participants also expressed concerns about loss of privacy through their participation in genetics research. Patients and parents in particular prioritized privacy in matters of family and health. They worried that genetics research could expose sensitive information about one's illness and family that they hoped to keep private. Previous studies cited a similar urge towards privacy as protection from stigma, oppression or harm [21,22,35].

## Study limitations

This study has some limitations that must be noted. Providers were recruited through convenience sampling and only those who were willing and able to set aside time to participate in the lengthy interview process took part in the study. Additionally, constraints introduced by the COVID-19 pandemic necessitated remote interviews. Therefore, only those individuals with the required hardware and software and a stable internet connection were able to participate in the study. The perspectives obtained may therefore not fully represent the overall views of Jamaican SCD stakeholders, but rather a subset with perhaps a greater openness to or means to participate scientific or clinical research. Relatedly, this study featured a greater number of female respondents compared to male respondents (72.4% vs 27.6%). This may have been due to several factors including the greater number of female providers and parents, the willingness or availability of male patients to participate in the study, among others. The study's largely female sample may have implications for the generalizability of results, particularly in the case of SCD patients.

## Practice implications

The data presented here highlight a strong willingness towards genetics research participation among Jamaican SCD stakeholders. Participants recognized the importance of diverse populations taking part in these studies. Researchers in genetics must take steps to facilitate this.

Genetics research in resource-constrained settings like Jamaica carries unique implications. Power imbalances are at the basis of many of these implications, particularly those that exist between foreign genetics researchers and host communities. Participants' misconceptions about the nature and purpose of genetics research (therapeutic and diagnostic misconceptions) can facilitate participant exploitation and diminish participant autonomy through impaired informed decision making. Greater communication and understanding of research goals and the implications of research participation will be essential for protecting participant autonomy throughout the research process. Additionally, trusted local researchers and providers will play an essential role in facilitating trustworthy research and establishing trust between researchers in genetics and participants, and overcoming concerns about misuse.

We present the following practice recommendations:

1) Implement community-sessions prior to study recruitment in which local clinicians, community health workers or trusted local researchers communicate research goals and processes and provide opportunities for questions in an informal setting.

2) Create culturally relevant tools and materials such as booklets or videos to better convey the risk, benefits, expectations, and limitations of genetics research participation, and facilitate greater transparency and improved trust.

3) Integrate Local Advisory Boards in Study Governance that includes local researchers, community advocates, patient representatives, and ethics committee members to advocate on behalf of participants and provide ongoing community feedback throughout the process to reduce power imbalances and facilitate local oversight.

4) Establish clear policies that dictate data use of and return of results that communicate to participants how their data will be stored, shared, and used, what results will be returned to participants, and the process by which these results will be returned to foster trust.

## Supporting information

**S1 Appendix. In-depth Interview Guides.**
(DOCX)

**S2 Appendix. Code Book.**
(DOCX)

**S1 Data. Minimal Data Set.**
(XLSX)

## Author contributions

**Conceptualization:** Krystin Jones, Kearsley Stewart, Monika Asnani, Charmaine D Royal.

**Formal analysis:** Krystin Jones.

**Investigation:** Krystin Jones.

**Methodology:** Krystin Jones.

**Supervision:** Kearsley Stewart, Monika Asnani, Charmaine D Royal.

**Writing – original draft:** Krystin Jones.

**Writing – review & editing:** Krystin Jones, Kearsley Stewart, Monika Asnani, Charmaine D Royal.

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
