## [Decision Letter · Decision Letter 0]

13 Oct 2025

PGPH-D-25-02453

Stakeholder Perspectives on Sickle Cell Disease Genetics Research in Jamaica

Dear Dr. Jones,

Thank you for submitting your manuscript to PLOS Global Public Health. After careful consideration, we feel that it has merit but does not fully meet PLOS Global Public Health’s publication criteria as it currently stands. Therefore, we invite you to submit a revised version of the manuscript that addresses the points raised during the review process.

We look forward to receiving your revised manuscript.

Kind regards,

Rahul Gajbhiye, MBBS PhD

Academic Editor

Journal Requirements:

1. Please send a completed 'Competing Interests' statement, including any COIs declared by your co-authors. If you have no competing interests to declare, please state "The authors have declared that no competing interests exist". Otherwise please declare all competing interests beginning with the statement "I have read the journal's policy and the authors of this manuscript have the following competing interests:"

1. Please clarify all sources of funding (financial or material support) for your study. List the grants (with grant number) or organizations (with url) that supported your study, including funding received from your institution.

2. State the initials, alongside each funding source, of each author to receive each grant.

3. State what role the funders took in the study. If the funders had no role in your study, please state: “The funders had no role in study design, data collection and analysis, decision to publish, or preparation of the manuscript.”

4. If any authors received a salary from any of your funders, please state which authors and which funders.

Additional Editor Comments (if provided):

Reviewer 1

The authors in this manuscript have described the perspectives of patients, family members and healthcare providers on role of genetics pertaining to SCD. My comments are as follows:

1. The title “Stakeholder Perspectives on Sickle Cell Disease Genetics Research in Jamaica” sounds broad but may overstate the range of perspectives included. Though the manuscript includes the perspectives of arguably the most directly affected stakeholders, the title doesn’t make this clear. A more specific title could show which groups are the focus, so readers know whose perspectives shape the analysis.

2. The framework which was used for thematic analysis should be included in the study design in the methods section.

3. Methods (Page 9, line number 162): Please clarify whether coding and thematic analysis were conducted by a single author or if codes and themes were independently reviewed/validated by a second author. Additionally, specify how data triangulation was ensured.

4. Results (Table 1): the median needs to be reported along with interquartile range instead of range of values.

5. Results (Pg 17, line 303-304): The statement “theme summarized in error, reference source not found!!!” requires clarification and correction.

Reviewer 2 In the submitted manuscript the authors have tried to understand the stakeholder’s perspectives about genetic research in sickle cell disease in Jamaica using qualitative research methods. As the possibilities for genetic treatment of sickle cell disease are growing, this becomes an important area to study.

I have the following major comment

• The study’s findings have important practice implications. In the last section of the manuscript the authors do provide some suggestion but they appear more abstract. The authors may want to provide 4 to 5 concrete suggestions based on their findings and may consider presenting these suggestions in a table. For example, the authors state- “Greater communication and understanding of research goals and the implications of research participation will be essential for protecting participant autonomy throughout the research process.” How can this be achieved? What would be mechanisms on the ground to ensure this?

There are some minor concerns-

• Why did the authors choose in-dept interviews over focus group discussions for patients and parents? Focus group discussions often bring out more information which more than the collective of the individual opinions. The authors may want to provide an explanation.

• The authors state that (lines 129, 130) “patients and parents were randomly selected from a list of patient data. A purposive sampling strategy was also used to ensure that male patients and parents were included in the study” These two sentences appear contrasting. Kindly provide further explanation.

• In qualitative studies, typically, mean and median data are not presented. The authors may want to just provide ranges in Table 1.

• In healthcare research the patients are often given least importance from a power perspective. In table 1 the authors present data on providers first followed by patients and parents. I do not know if the authors had something specific in mind while arranging the data in this way but I would suggest presenting data on patients and parents first followed by those from the providers.

• This is an extension of my first comment. Most of the responders in the study were women which is welcome. The readers may want to know if a majority of patients attending the SCD clinic were women. Also, will this overrepresentation have implications for the findings of the study? Authors may want to discuss this in the manuscript.

• There is a typo on line 303 which the authors should correct. “in Error! Reference source not found”

Reviewers' comments:

Reviewer's Responses to Questions

**Comments to the Author**

1. Does this manuscript meet PLOS Global Public Health’s publication criteria?

Reviewer #1: Partly

Reviewer #2: Yes

2. Has the statistical analysis been performed appropriately and rigorously?

Reviewer #1: N/A

Reviewer #2: N/A

3. Have the authors made all data underlying the findings in their manuscript fully available (please refer to the Data Availability Statement at the start of the manuscript PDF file)?

Reviewer #1: No

Reviewer #2: No

4. Is the manuscript presented in an intelligible fashion and written in standard English?

Reviewer #1: Yes

Reviewer #2: Yes

Reviewer #1: The authors in this manuscript have described the perspectives of patients, family members and healthcare providers on role of genetics pertaining to SCD. My comments are as follows:

1. The title “Stakeholder Perspectives on Sickle Cell Disease Genetics Research in Jamaica” sounds broad but may overstate the range of perspectives included. Though the manuscript includes the perspectives of arguably the most directly affected stakeholders, the title doesn’t make this clear. A more specific title could show which groups are the focus, so readers know whose perspectives shape the analysis.

2. The framework which was used for thematic analysis should be included in the study design in the methods section.

3. Methods (Page 9, line number 162): Please clarify whether coding and thematic analysis were conducted by a single author or if codes and themes were independently reviewed/validated by a second author. Additionally, specify how data triangulation was ensured.

4. Results (Table 1): the median needs to be reported along with interquartile range instead of range of values.

5. Results (Pg 17, line 303-304): The statement “theme summarized in error, reference source not found!!!” requires clarification and correction.

Reviewer #2: In the submitted manuscript the authors have tried to understand the stakeholder’s perspectives about genetic research in sickle cell disease in Jamaica using qualitative research methods. As the possibilities for genetic treatment of sickle cell disease are growing, this becomes an important area to study.

I have the following major comment

• The study’s findings have important practice implications. In the last section of the manuscript the authors do provide some suggestion but they appear more abstract. The authors may want to provide 4 to 5 concrete suggestions based on their findings and may consider presenting these suggestions in a table. For example, the authors state- “Greater communication and understanding of research goals and the implications of research participation will be essential for protecting participant autonomy throughout the research process.” How can this be achieved? What would be mechanisms on the ground to ensure this?

There are some minor concerns-

• Why did the authors choose in-dept interviews over focus group discussions for patients and parents? Focus group discussions often bring out more information which more than the collective of the individual opinions. The authors may want to provide an explanation.

• The authors state that (lines 129, 130) “patients and parents were randomly selected from a list of patient data. A purposive sampling strategy was also used to ensure that male patients and parents were included in the study” These two sentences appear contrasting. Kindly provide further explanation.

• In qualitative studies, typically, mean and median data are not presented. The authors may want to just provide ranges in Table 1.

• In healthcare research the patients are often given least importance from a power perspective. In table 1 the authors present data on providers first followed by patients and parents. I do not know if the authors had something specific in mind while arranging the data in this way but I would suggest presenting data on patients and parents first followed by those from the providers.

• This is an extension of my first comment. Most of the responders in the study were women which is welcome. The readers may want to know if a majority of patients attending the SCD clinic were women. Also, will this overrepresentation have implications for the findings of the study? Authors may want to discuss this in the manuscript.

• There is a typo on line 303 which the authors should correct. “in Error! Reference source not found”

**Do you want your identity to be public for this peer review?** For information about this choice, including consent withdrawal, please see our Privacy Policy

Reviewer #1: No

Reviewer #2: No

---

## [Decision Letter · Decision Letter 1]

19 Dec 2025

Patient, Parent and Provider Perspectives on Sickle Cell Disease Genetics Research in Jamaica

PGPH-D-25-02453R1

Dear Dr Jones,

We are pleased to inform you that your manuscript 'Patient, Parent and Provider Perspectives on Sickle Cell Disease Genetics Research in Jamaica' has been provisionally accepted for publication in PLOS Global Public Health.

Best regards,

Rahul Gajbhiye, MBBS PhD

Academic Editor

The authors may want to revisit the median and range/interquartile range for patient and provider (needful has been done for parent).

All other comments  addressed.

Reviewer Comments (if any, and for reference):

Reviewer's Responses to Questions

**Comments to the Author**

Reviewer #1: All comments have been addressed

Reviewer #2: All comments have been addressed

publication criteria?

Reviewer #1: Yes

Reviewer #2: Yes

3. Has the statistical analysis been performed appropriately and rigorously?

Reviewer #1: N/A

Reviewer #2: N/A

4. Have the authors made all data underlying the findings in their manuscript fully available (please refer to the Data Availability Statement at the start of the manuscript PDF file)?

Reviewer #1: Yes

Reviewer #2: Yes

5. Is the manuscript presented in an intelligible fashion and written in standard English?

Reviewer #1: Yes

Reviewer #2: Yes

Reviewer #1: The authors may want to revisit the median and range/interquartile range for patient and provider (needful has been done for parent).

Reviewer #2: (No Response)

**Do you want your identity to be public for this peer review?** For information about this choice, including consent withdrawal, please see our Privacy Policy

Reviewer #1: No

Reviewer #2: No
